# Origin of Subsequent Malignant Neoplasms in Patients with History of Testicular Germ Cell Tumor

**DOI:** 10.3390/cancers12123755

**Published:** 2020-12-14

**Authors:** Eric C. Umbreit, Bilal A. Siddiqui, Michael J. Hwang, Aron Y. Joon, Tapati Maity, Mary E. Westerman, Kelly W. Merriman, Hussam Alhasson, Joma Uthup, Tao Guo, Joseph A. Moore, John F. Ward, Jose A. Karam, Christopher G. Wood, Louis L. Pisters, Miao Zhang, Shi-Ming Tu

**Affiliations:** 1Department of Urology, University of Texas MD Anderson Cancer Center, Houston, TX 77030, USA; umbreite@gmail.com (E.C.U.); TMaity@mdanderson.org (T.M.); MEWesterman@mdanderson.org (M.E.W.); JFWard@mdanderson.org (J.F.W.); JAKaram@mdanderson.org (J.A.K.); CGWood@mdanderson.org (C.G.W.); lpisters@mdanderson.org (L.L.P.); 2Department of Genitourinary Medical Oncology, University of Texas MD Anderson Cancer Center, Houston, TX 77030, USA; basiddiqui@mdanderson.org (B.A.S.); hussam.alhasson@gmail.com (H.A.); JUthup1@mdanderson.org (J.U.); JAMoore6@mdanderson.org (J.A.M.); 3Department of Pathology, University of Texas MD Anderson Cancer Center, Houston, TX 77030, USA; MJHwang@mdanderson.org (M.J.H.); TGuo@mdanderson.org (T.G.); MZhang8@mdanderson.org (M.Z.); 4Department of Biostatistics, University of Texas MD Anderson Cancer Center, Houston, TX 77030, USA; AYJoon@mdanderson.org; 5Department of Tumor Registry, University of Texas MD Anderson Cancer Center, Houston, TX 77030, USA; kmerriman@mdanderson.org

**Keywords:** testicular cancer, second malignancy, somatic transformation, isochromosome 12p, teratoma, cancer stem cell

## Abstract

**Simple Summary:**

Although testicular germ cell tumor (TGCT) carries a high cure rate, some patients still die from it. We investigated the genetic landscape and cellular origins of cancers that develop later in life after treatment for TGCT and found evidence that a common progenitor cell might be responsible for both. This study shows the possible importance of stem-like cells in the development of cancer.

**Abstract:**

Although genetic changes may be pivotal in the origin of cancer, cellular context is paramount. This is particularly relevant in a progenitor germ cell tumor and its differentiated mature teratoma counterpart when it concerns tumor heterogeneity and cancer dormancy in subsequent second malignancies (subsequent malignant neoplasms (SMNs)). From our tumor registry database, we identified 655 testicular germ cell tumor (TGCT) patients who developed SMNs between January 1990 and September 2018. Of the 113 solid organ SMNs, 42 had sufficient tumor tissue available for fluorescence in situ hybridization (FISH) analysis of isochromosome 12p [i(12p)]. We identified seven additional patients for targeted DNA and RNA sequencing of teratomas and adjacent somatic transformation. Finally, we established cell lines from freshly resected post-chemotherapy teratomas and evaluated the cells for stemness expression by flow cytometry and by the formation of teratomas in a xenograft model. In our cohort, SMNs comprising non-germ cell tumors occurred about 18 years after a diagnosis of TGCT. Of the 42 SMNs examined, 5 (12%) contained i(12p) and 16 (38%) had 12p gain. When comparing a teratoma and adjacent somatic transformation, targeted DNA and RNA sequencing demonstrated high concordance. Studies of post-chemotherapy teratoma-derived cell lines revealed cancer-initiating cells expressing multipotency as well as early differentiation markers. For the first time, we demonstrated the prevalence of i(12p) in SMNs and the presence of progenitor cells embedded within mature teratomas after chemotherapy. Our findings suggest a progenitor stem-like cell of origin in SMN and TGCT and highlight the importance of cellular context in this disease.

## 1. Introduction

Although testicular germ cell tumor (TGCT) carries a high cure rate (exceeding 90%), a subset of patients still succumb to it [1,2,3]. Identification of patients at risk for a lethal TGCT remains a key unmet need. Mortality from TGCTs can occur in part due to the somatic transformation of teratomas into non-TGCT subtypes (e.g., sarcoma, primitive neuroectodermal tumor, or adenocarcinoma) or the development of subsequent malignant neoplasms (SMNs). Somatic transformation occurs in 2.4% of primary nonseminomatous germ cell tumors (NSGCTs) and 3.4% of residual metastatic NSGCTs after chemotherapy [3]. Mortality from somatic transformation appears to be driven by transformation in metastatic lesions (with a mortality rate of 68%), rather than transformation of the primary testicular tumor [3].

Groot et al. reported that the cumulative incidence of SMNs is about 10% over the 25-year period following treatment of TGCT [4]. They attributed SMNs to the accumulation of deleterious mutations related to DNA damage from prior chemotherapy and/or radiotherapy, as well as to the process of somatic transformation [4]. Both somatically transformed tumors and TGCTs contain the isochromosome 12p [i(12p)] [5], but it is unclear whether SMNs also harbor i(12p). Additionally, a pure seminoma (which does not classically undergo somatic transformation) may also evolve into an i(12p)-containing SMN without prior exposure to chemotherapy or radiotherapy, highlighting the role of lineage plasticity and cellular context in the development of SMNs [6,7,8,9,10].

We sought to use i(12p) as a genetic marker to determine if an SMN represents a new primary malignancy or is related to a prior primary TGCT. Approximately 80% of TGCTs harbor i(12p); the remaining 20% carry other chromosomal changes involving chromosome 12p, including increases in copy number, tandem duplication, or transposition [11,12,13,14]. The consistent involvement of 12p may implicate common gene elements located on that chromosome, including C13-16CND2, SOX5, JAW1, KRAS, NANOG, and GDF3, in the pathogenesis of TGCT [14,15,16]. Importantly, i(12p) is almost never associated with other malignancies without previous or concurrent TGCT [17]. However, 12p by itself is insufficient for the development of TGCT—in Pallister–Killian mosaic syndrome [18], patients with i(12p) or tetrasomy 12p do not develop TGCT. Therefore, while i(12p) does not necessarily produce TGCT, its high prevalence in TGCT makes it a useful genetic marker to establish potential links between TGCT and SMN.

In this study, we investigated the cellular origin of SMN, somatically transformed teratoma, and surgically excised post-chemotherapy teratoma. We evaluated the prevalence of i(12p) (and gain of 12p) by fluorescence in situ hybridization (FISH) on various SMNs in patients with a history of TGCT. In addition, we performed targeted DNA (T200) and RNA sequencing (RNAseq) of matched metastatic teratomas and somatically transformed tumors from seven patients. Finally, we established cell lines from post-chemotherapy residual teratomas of two patients and identified progenitor cell populations by flow cytometry and the formation of teratomas in xenograft. Our findings suggest that an early progenitor cell in TGCT drives lethal phenotypes, including SMN and somatic transformation, highlighting the importance of cellular context in our understanding of this disease.

## 2. Results

### 2.1. Detection of i(12p) and Gain of 12p in SMNs

The study cohort comprised 43 SMNs from 42 patients (Table 1). Two separate patients with SMNs positive for i(12p) underwent testing of normal adjacent tissue without cancer as a negative control. We did not detect i(12p) or 12p gain in the normal tissues. The median age at the time of diagnosis of primary TGCT was 41 years (range: 20–68). The median age at the time of diagnosis of SMN was 59 years (range: 34–86). The median time to diagnosis of SMN from diagnosis of primary TGCT was 18 years (range: 0–57). From the 42 cases of SMNs, 21 (50%) were found to carry either i(12p) (*n* = 5, 12%) or additional copies of chromosome 12 without apparent gain of 12p (*n* = 16, 38%). The clinicopathologic characteristics and FISH results of these SMNs are listed in Table 2. Representative images demonstrating detection of i(12p) or additional copies of chromosome 12 without i(12p) are shown in Figure 1Aand 1B respectively. The negative control is depicted in Figure 1C. FISH data by tumor site are provided in Appendix A.

### 2.2. Evaluation of Genetic Concordance in the Somatic Transformation of TGCTs by Targeted DNA Sequencing

To better characterize the genetic background of somatically transformed teratoma, we identified seven patients who underwent surgical resection of a tumor with both a teratoma and somatic transformation present at a metastatic site, providing matched specimens. The clinical characteristics of these patients and the chemotherapy regimens they received are shown in Appendix A. Using the T200 gene panel, we detected mutations in 58 genes across all 14 samples [19] (Figure 2). The McNemar exact test yielded no evidence of pair discordance on all genes, including 24 genes with slight discordance and others with near-perfect concordance between the teratoma and the paired somatic transformation. Among the 58 genes with mutations, the “within-patient mutation agreement” ranged from 86.2% to 94.8%, indicating a high degree of concordance among the 200 genes assessed (Figure 2, Appendix A).

### 2.3. Evaluation of Differentially Expressed Genes in Matched Teratoma and Somatically Transformed Specimens by RNAseq

We next evaluated expressed genes in our matched teratomas and somatically transformed tumors using RNAseq. We similarly observed a high degree of concordance across a majority of genes. Notably, we identified seven genes with significantly lower expression in somatically transformed tissue: *miR-3156-1, miR-7-3HG, ARHGEF35, LZTS1-AS1, LIPN, AC243960.3*, and *DLX6* (Appendix A). The maximum log2 fold change gene expression was moderate (2.04).

### 2.4. Growth of Teratoma Cell Cultures and Xenograft

Cell cultures were propagated from post-chemotherapy teratoma samples from two patients (Appendix A), creating cell lines Tera13 and Tera22. As described in mouse teratocarcinoma cell cultures [20], classic “C” cell colonies formed in STEMium media (Figure 3A). For controls, samples of resections obtained from patients with benign post-chemotherapy masses, which contained fibrosis and necrosis, were processed identically (at the time of resection, all masses had unknown final pathology). The samples generated poorly growing colonies or no cell colonies at all. There were no “C” cell colonies identified. “C” and “F” cell colonies were able to be grown in DMEM (Appendix A). Finally, the Tera13 cell line was able to successfully recapitulate teratoma formation in xenograft models (Figure 3B, Appendix A).

### 2.5. Stemness Biomarker Expression

When tested by flow cytometry, early-passage cells expressed classic cancer-initiating cell surface markers, suggesting the stem-like and multipotent nature of these cells (Table 3) [21,22]. Approximately 50% of the cells expressed SSEA-4, a marker of normal adult spermatogonial stem cells. There was presence of CD34 (3%) but loss of SSEA5, [23] TRA-1-60, TRA-1-81, and Cripto-1. We performed Western blots and detected NANOG but not OCT3/4 or SALL4 in Tera13, and no NANOG, OCT3/4, or SALL4 in Tera22 (Appendix A). Uncropped Western blots are shown in Appendix A.

## 3. Discussion

In this study, we detected the presence of i(12p) or gain of 12p in half of the patients with SMN and a previous diagnosis of TGCT. By targeted DNA sequencing and RNAseq, we identified high genetic concordance (86–95%) in patient-matched teratomas and somatically transformed tumors, suggesting a common clonal origin of these two clinically and pathologically distinct tumor types. We successfully established post-chemotherapy teratoma cell cultures and demonstrated the existence of progenitor cells by their expression of stemness surface markers in vitro and their exhibition of multipotency in vivo. Our findings suggest that progenitor stem-like cells of origin may account for SMN and somatic transformation, which ultimately determine the lethality of TGCT.

In our FISH studies, we did not detect i(12p) or 12p gain in normal control tissues, confirming that this was a somatic rather than germline genetic aberration. According to Groot et al., the cumulative incidence of SMN after treatment of TGCT was similar for patients with a history of seminoma or nonseminoma (about 12.5% and 9.5% over a 25-year period, respectively). Radiation therapy and platinum-containing chemotherapy were associated with an increased risk of solid SMN, particularly with GI SMNs in a dose-dependent manner [4]. Similar to previous studies [4,24,25], we found that SMN with positive i(12p) or 12p gain could occur in a variety of organ sites. We also found that a preponderance of SMNs had occurred in patients with a history of seminoma and radiation therapy [26], and that a majority of those SMNs had actually occurred outside the irradiated field. Interestingly, while almost all patients had prior radiation therapy or chemotherapy for their TGCT treatment, one patient (#14, Table 1), who developed an SMN in the lung, had neither.

Therefore, the development of SMN in patients with a history of TGCT is not entirely explained by DNA damage and genomic aberrations from toxic curative therapy. Interestingly, progenitor stem-like cells in the phenomenon of microchimerism can migrate to distant sites, differentiate into various cellular lineages, and remain dormant for a prolonged period. Hence, progenitor stem-like cells in fetal-to-maternal microchimerism migrate from the fetus to the mother across the placenta and form hybrid tissues within the mother. Microchimerism has been detected as early as seven weeks into pregnancy and as late as 27 years postpartum [27,28,29,30]. Similarly, progenitor stem-like cells in donor microchimerism migrate between individuals after solid organ transplantation. In cases of malignancy following solid organ transplantation, about 12% originate from the donor organ, while 8% of the malignancies in the donated organ have a recipient origin [31]. Because progenitor stem-like cells migrate and may transform in different individuals, it is conceivable that they may also do so within the same individual.

Intriguingly, the incidence of solid SMNs has remained more or less unchanged from 1980 to 2008, despite overall deintensification of therapy [31,32,33,34,35,36], which suggests that DNA-damaging chemotherapy and radiotherapy may not fully explain SMNs and raises the hypothesis that a common cell of origin may be responsible for subsequent lethal phenotypes. A mature teratoma provides an ideal opportunity for testing this hypothesis, because (1) it is a fully differentiated tumor linked to late relapses (as in SMN) and to somatic transformation, if not removed in a timely manner; and (2) it often coexists with somatically transformed malignancies, allowing investigation of their shared genetic signature and common clonal origin.

Currently, the prevailing hypothesis suggests that indigenous differentiated cells dedifferentiate and acquire stemness properties, that is, undergo transformation or reprogramming by way of genetic aberrations or mutations [37,38]. We sought to test this hypothesis in mature teratomas—a fully differentiated tumor that often coexists with somatically transformed malignancies within the same tumor in the same patient. We found that metastatic teratomas and their corresponding somatic transformations displayed highly concordant genetic profiles based on genome sequencing, which points to a common clonal origin and suggests that specific genetic aberrations are not responsible for dedifferentiation or somatic transformation.

Our results support a stem-cell origin of cancer and validate the importance of cellular context in TGCT patients who develop SMN [6,7]. Late TGCT recurrences and SMNs developing many years—even decades—later do not appear to be due to the acquisition and accumulation of new genetic mutations, but rather to the expression or revelation of aberrant stem-like or embryonic epigenetic phenotypes. We postulate that even minimal or minor damage to a pertinent malignant cell of origin (i.e., progenitor stem-like cells rather than progeny differentiated cells) triggers an SMN at a distant site in a distant time.

Finally, accounting for cellular context has key therapeutic implications. After all, a progenitor chemo-sensitive TGCT (e.g., an embryonal carcinoma) and a progeny chemo-resistant TGCT (e.g., a teratoma) harbor the same genetic defects due to their common clonal origins but have different epigenetic phenotypes (e.g., migratory prowess, tumor heterogeneity, and cancer dormancy) [39,40]. Furthermore, the presence of a teratoma [41] in the primary tumor suggests a potential lethal tumor phenotype, intrinsically refractory to chemotherapy and prone to somatic transformation, that may require an alternative therapeutic regime or strategy and can only be cured by surgical resection in a timely manner [3,42].

A limitation of this study relates to rarity of disease and paucity of clinical samples. We need to perform targeted DNA sequencing and RNASeq on matched patient specimens to compare primary tumors with SMNs as well as among SMNs in patients with more than one SMN. We also need to demonstrate that the presence and type of progenitor cells within teratomas may vary depending on the type of primary TGCT from which they arise [3,42]. However, we hope that the raised awareness from this study will spur expanded profiling on greater numbers of pertinent tumors.

## 4. Materials and Methods

### 4.1. Patient Identification

We retrospectively analyzed the tumor registry at the MD Anderson Cancer Center (MDACC) and identified 655 TGCT patients who developed a subsequent SMN between January 1990 and September 2018. We focused on 113 recent cases (since 2011) of solid tumor SMNs, of which 42 had sufficient tissue to perform fluorescence in situ hybridization (FISH) of i(12p). We also identified 7 cases between 2001 and 2008 in which the teratoma and the somatic transformation were adjacent within the same residual tumor after chemotherapy with sufficient tissue for targeted DNA and RNA sequencing.

### 4.2. Fluorescent in Situ Hybridization (FISH)

FISH was performed on unstained slides prepared from archival tissue at the Mayo Clinic Cytogenetics Core to detect the presence of i(12p). FISH was also performed on 1 test case at the MDACC Cytogenetics Laboratory for confirmation of the results and with matched normal tissue for a negative control and the detection of germline i(12p). In brief, the working solution contained 3 μL concentrated PKP2 DNA probe labeled with SpectrumOrange and 1 μL of concentrated D12Z3 probe labeled with SpectrumGreen (Abbott Molecular/Vysis Products, Abbott Park, IL, USA) in 16 μL of LSI/WCP hybridization buffer (Abbott Laboratories). Two technologists independently scored 50 qualifying tumor nuclei for each sample. In the event of a discrepancy, a third technologist would also score the sample and report the original score closest to its value. The normal cutoff value for enumeration of the i(12p) probe set with 95% (*p* < 0.05) confidence level was 30% [43].

### 4.3. DNA and RNA Sequencing

Details of RNA and cDNA library preparation and capture are provided in Appendix A. The captured libraries were sequenced on a HiSeq 2500 (Illumina Inc., San Diego, CA, USA) on a version 3 TruSeq paired-end flow cell according to the manufacturer’s instructions at a cluster density between 700 and 1000 K clusters/mm^2^. Density is measured as 1000 (K) clusters per square millimeter (mm²). Sequencing was performed for 2 × 100 paired-end reads with a 7 nt read for indexes using cycle sequencing v3 reagents (Illumina). The resulting BCL files containing the sequence data were converted into FASTQ files, and individual libraries within the samples were de-multiplexed using CASAVA 1.8.2 with no mismatches. All regions were covered by >20 reads.

### 4.4. Primary Culture, Flow Cytometry, and Xenograft

We procured and propagated post-chemotherapy teratoma cells, as described in the Appendix A. In short, tumor cells were incubated at 37 °C in 10% CO_2_ and 5% O_2_ in STEMium—a human pluripotent stem cell growth medium (ScienCell, Carlsbad, CA, USA). The culture medium was changed every 2 days and the cells passed as necessary with 80% to 90% confluence. The final pathology of teratoma in the patient was confirmed within 7 days. Cell line samples were stained live for cell-surface stemness markers expressed on hESCs and cancer stem cells. Details of flow cytometry analysis are provided in the Appendix A [20]. Early passages were injected in SCID mice and are further described in the Appendix A.

### 4.5. DNA Sequencing Analysis

The FASTQ data from Illumina HiSeq were processed at the Institute for Personalized Cancer Therapy (IPCT) at MD Anderson through standard pipelines outlined at https://www.ncbi.nlm.nih.gov/pmc/articles/PMC4511273/. In essence, the target-capture deep-sequencing data were aligned to human reference assembly hg19 using BWA, and duplicated reads were removed using Picard. The single nucleotide variants (SNVs) and small indels were called by an in-house analysis pipeline, which classified variants into 3 categories, somatic, germline, and loss of heterozygosity, based on variant allele frequencies in the tumor and the matched normal tissues.

### 4.6. RNA Sequencing Analysis

The FASTQ data were processed by an in-house RNAseq data analysis pipeline, which used the STAR aligner [44] to align raw reads to the hg19 human reference genome, followed by Oncofuse [45] to filter and prioritize fusion candidates, then the use of featureCounts [46] to quantify aligned reads to produce raw counts, and FastQC and Qualimap [47] to evaluate the quality of raw reads and feature counts. Hierarchical clustering and k-means unsupervised clustering algorithms were used to cluster samples according to their RNAseq expression profiles. The details of the sequencing analysis and interpretation are provided in the Appendix A.

### 4.7. Concordance of Mutations

To assess the within-subject mutation concordance/discordance between teratoma and somatic transformation samples, we conducted the McNemar exact test against the null hypothesis of concordance (i.e., the equal proportion of discordant pairs) on each of 58 genes on the T200 panel with detected mutation.

## 5. Conclusions

In summary, we detected i(12p) or gain of 12p in a substantial number of SMNs among patients with a history of TGCT and demonstrated high genetic concordance between their teratoma and adjacent somatically transformed tumor, with few differentially expressed genes, suggesting that specific genetic aberrations are not major drivers of somatic transformation. We discovered putative progenitor cells embedded within mature teratomas and were able to recapitulate teratoma formation in one case of a xenograft cell line, suggesting that progenitor cells differentiate into diverse cellular lineages [48,49] rather than progeny cells dedifferentiating into stem-like cells through genetic mutations. Further studies in additional patients are warranted to confirm whether a progenitor stem-like cell of origin could account for both SMN development and somatic transformation in patients with a history of TGCT.

## Figures and Tables

**Figure 1 cancers-12-03755-f001:**
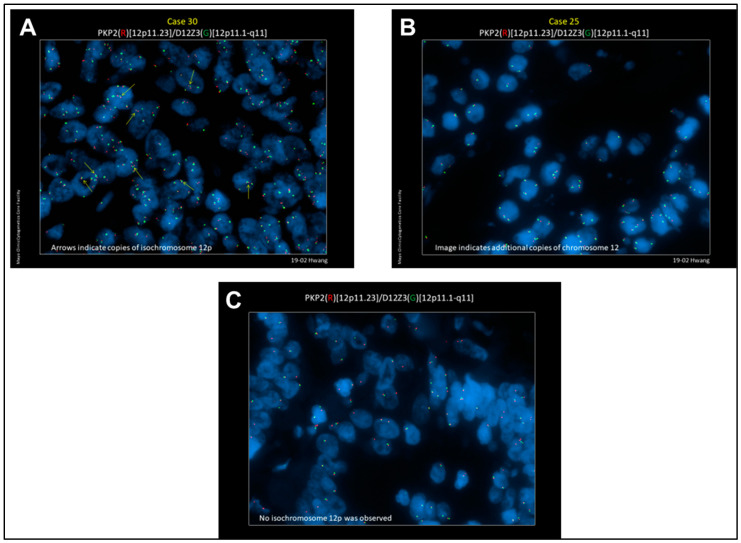
Representative FISH images. (**A**) Representative image (patient #4) demonstrating detection of i(12p) by FISH using PKP2 DNA probe labeled with SpectrumOrange and D12Z3 probe labeled with SpectrumGreen (see text for details). (**B**) Representative image (patient #13) demonstrating additional copies of chromosome 12 without detection of i(12p). (**C**) Representative image demonstrating a negative control.

**Figure 2 cancers-12-03755-f002:**
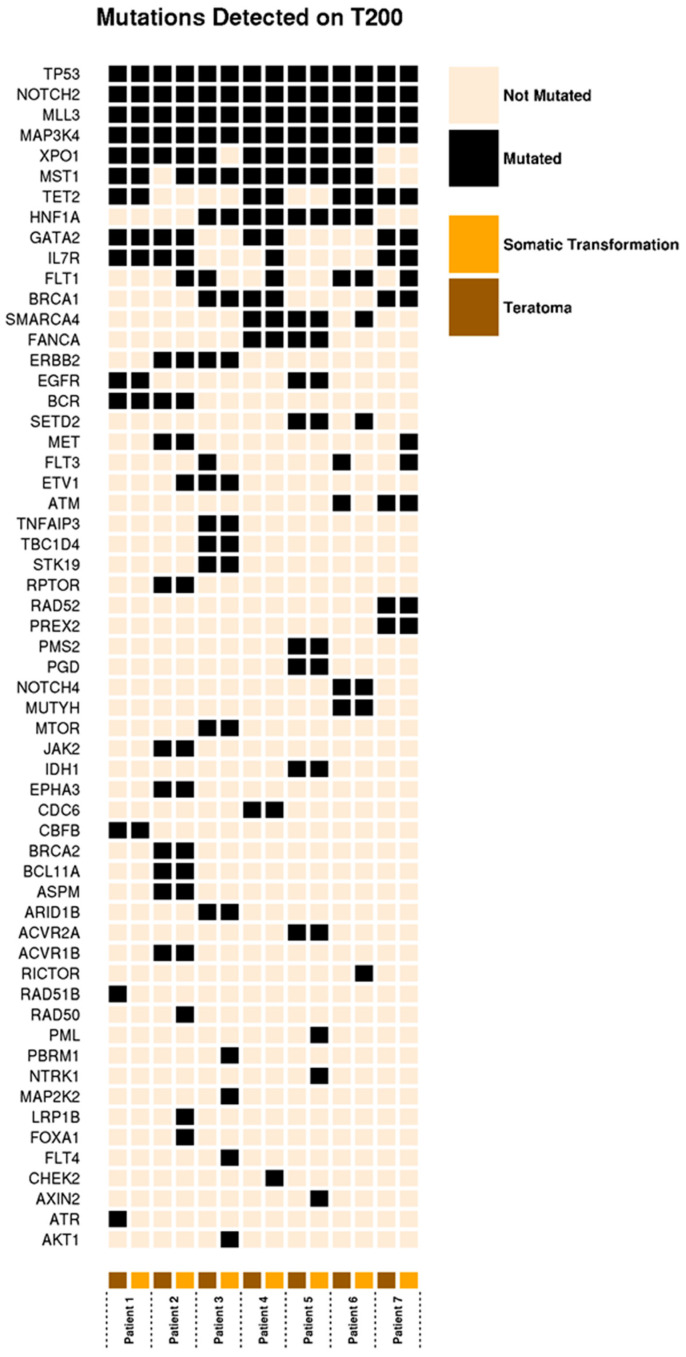
OncoPrint data of mutations detected in 14 samples across 7 patients.

**Figure 3 cancers-12-03755-f003:**
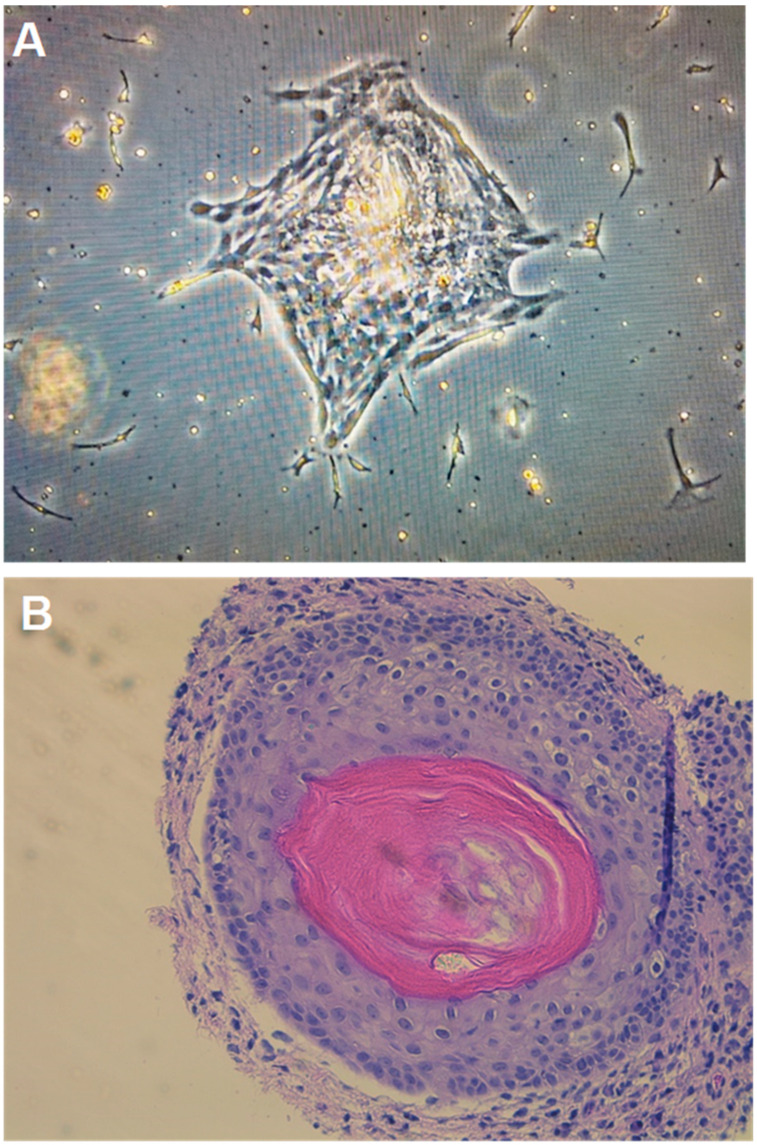
(**A**) Representative image of Tera13 growing in STEMium at passage 0, day 9. (**B**) Teratoma with chondrocyte and lamellar bone formation in a xenograft model. Tera13 cell line derived from primary culture of a residual mature teratoma following chemotherapy.

**Table 1 cancers-12-03755-t001:** Distribution of subsequent malignant neoplasms (SMNs) in testicular germ cell tumor (TGCT) cohort.

Second Malignancy	Number of Cases
**Total**	**43**
**Gastrointestinal**	**9**
Colorectal	6
Pancreas	2
Adenocarcinoma	1
Neuroendocrine	1
Esophageal	1
**Lung**	**7**
Squamous	3
Adenocarcinoma	3
Neuroendocrine	1
**Genitourinary**	**15**
Renal cell	8
Clear cell	7
Papillary	1
Urothelial	5
Bladder	4
Upper tract	1
Prostate	2
**Sarcoma**	**7**
Leiomyosarcoma	4
Retroperitoneal	3
Gluteal	1
Rhabdomyosarcoma	1
Angiosarcoma	1
Unclassified	1
**Melanoma**	**3**
**Glioblastoma multiforme**	**1**
**Adrenal cortical carcinoma**	**1**

**Table 2 cancers-12-03755-t002:** Detection of i(12p) or additional copies of chromosome 12 without i(12p) by fluorescence in situ hybridization (FISH) in primary tumors and SMNs.

**Case No.**	**Primary Testicular Tumor**	**Age at Initial Diagnosis**	**Primary Treatment**	**Age at SMN**	**Time to SMN (Yrs.)**	**SMN Histology**	**Metastatic Tumor**	**% i(12p) FISH Positive**	**# i(12p) Copies**
1	Seminoma *	51	BEP	57	6.5	Unclassified sarcoma	RP	56%	1–2
2	Seminoma	31	XRT	54	22.0	Colorectal	Liver	68%	1
3	Seminoma	28	XRT, BEP, HDC + SCT	34	5.9	LMS	Gluteus	64%	1–2
4	Seminoma	53	BEP	56	3.1	Urothelial	Bladder	39%	1
5	Nonseminoma	39	BEP	39	0.4	Pancreatic	Pancreas	63%	1–3
**Case No.**	**Primary Histology**	**Age at Initial Diagnosis**	**Primary Treatment**	**Age at SMN**	**Time to SMN (Yrs.)**	**SMN Histology**	**Site of SMN**	**% Extra Copies Chromosome 12**	**# i(12p) Copies**
6	Seminoma	33	XRT	51	18.0	LMS	RP	73%	0
7	Seminoma	51	XRT	59	7.6	Rectal (adenocarcinoma)	Rectum	55%	0
8	Seminoma	58	XRT	71	13.3	Colorectal	Cecum	36%	0
9	Seminoma	57	XRT	65	8.5	Lung (adenocarcinoma)	Lung	72%	0
10	Seminoma	29	XRT	59	24.6	Melanoma	Axillary lymph node	83%	0
	Nonseminoma	44	BEP						
11	Seminoma	22	XRT	58	9.6	LMS	RP	72%	0
12	Nonseminoma	28	CISCA/VB	45	36.3	LMS	RP	74%	0
13	Seminoma	53	XRT	58	44.8	Colorectal	Lung	83%	0
14	Seminoma	62		65	5.2	Lung (squamous)	Lung	61%	0
15	Seminoma	27	XRT	81	3.0	Urothelial	Bladder	66%	0
16	NA	20	XRT + VB	57	53.6	Adrenal Cortical Carcinoma	Adrenal	89%	0
17	Seminoma	31	XRT	61	36.6	Bladder	Bladder	82%	0
18	Seminoma	29	CTX/carboplatin	49	29.9	Urothelial	Renal pelvis	51%	0
19	Seminoma	47	XRT	65	20.3	Renal cell carcinoma	Kidney	57%	0
20	Seminoma	29	PVB	54	19.4	Urothelial	Bladder	91%	0

BEP: bleomycin, etoposide, cisplatin. XRT: radiation therapy. HDC + SCT: high-dose chemotherapy + stem-cell transplantation. CEB: carboplatin, etoposide, bleomycin. PVB: cisplatin, vinblastine, bleomycin. CISCA: cyclophosphamide, doxorubicin, cisplatin. VB: vinblastine, bleomycin. CTX: cyclophosphamide. LMS: leiomyosarcoma. RP: retroperitoneum. * Diagnosed by biopsy of retroperitoneal lymph node. Pathology of primary tumor not available.

**Table 3 cancers-12-03755-t003:** Tera13 and Tera22 expressed cell surface markers by flow cytometry, suggesting their stem-like and multipotent nature.

CSC Surface Marker	Function	Adult Stem Cell Expression	Normal Tissue Expression	CSC Expression	% Cells Positive in Cell Line tera13	% Cells Positive in Cell Line tera22
SSEA3	hESC marker	Mesenchymal	Rare	Embryonal, Breast	<1%	<1%
SSEA4	hESC marker	Mesenchymal, Cardiac, Gonad	Rare	Embryonal, Breast	55%	55%
SSEA5	hESC marker	NA	Unknown	Embryonal, Pancreatic, Colorectal, Urothelial	<1%	<1%
TRA-1-60	hESC marker	NA	Rare	Embryonal, Breast, Prostate	--	--
TRA-1-81	hESC marker	NA	Rare	Embryonal, Breast	--	--
CD133	Marker of hematopoietic stem cell	Hematopoietic, Prostate, Neural	Rare	Many *	5%	<1%
CD90	Cell adhesion	Mesenchymal, Cardiac	Rare	Brain, Liver	>99%	>99%
CD326	Cell adhesion	No	Rare	Colon, Pancreas, Liver	--	--
Cripto-1 (TDGF1)	Self-renewal	NA	Rare	Breast, Colon, Lung	4%	5%
CD24	B cell proliferation	Intestinal	Rare	Breast, Gastric, Pancreas	25%	2%
CD117 (c-kit)	Receptor for stem cell factor	Mesenchymal, Cardiac	Rare	Ovary	2%	--
CD26	Dipeptidyl peptidase iv	Hematopoietic	Rare	Colorectal, Leukemia	80%	>99%
CD34	Cell adhesion	Hematopoietic	Rare	Leukemia, SCC	3%	2%
CD44	Hyaluronic acid receptor	Hematopoietic, Adipose	Rare	Many **	>99%	>99%

CSC: cancer stem cell. hESC: human embryonic stem cell. NA: not applicable. SCC: squamous cell carcinoma. * Breast, prostate, colorectal, glioma, liver, lung, ovary; ** Squamous cell carcinoma, breast, colon, liver, ovary, pancreas, gastric.

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
