# Peer review of "Origin of Subsequent Malignant Neoplasms in Patients with History of Testicular Germ Cell Tumor"

_cancers, 2020, doi:10.3390/cancers12123755_

Round 1

Reviewer 1 Report

The authors should be commented on their pre-clinical work in teratoma, a subtype of germ cell tumors (GCT) poorly studied and understood and on their successful establishment of a pre-clinical xenograft model of teratoma.

However, this manuscript does not add much to the current body of knowledge of germ cell tumor biology and does not provide enough data to explain the secondary malignancy tumorigenesis in GCT survivors.

I found the manuscript confusing. While the primary objective is to demonstrate a common cell progenitor in SMN, the translational studies have been conducted using transformed teratoma, inferring that teratoma:transformed teratoma=primary GCT:SMN.

  1. The authors need to prove this correlation to use the transformed teratoma model as SMN. This assumption is not based on solid scientific evidence
  2. Several studies have demonstrated the association between active treatments, RT and chemotherapy, and development of SMN in GCT patients. To demonstrate that SMN are arising from GCT cells the authors should analyze a cohort of GCT patients who developed SMN and were not previously exposed to neither chemotehrapy or RT.

Overall, the data presented do not support the authors' conclusion that SMN originate from a common stem-cell from the original GCT.

Author Response

Reviewer #1

The authors should be commented on their pre-clinical work in teratoma, a subtype of germ cell tumors (GCT) poorly studied and understood and on their successful establishment of a pre-clinical xenograft model of teratoma.

However, this manuscript does not add much to the current body of knowledge of germ cell tumor biology and does not provide enough data to explain the secondary malignancy tumorigenesis in GCT survivors.

The finding of i(12p)/12p gain in SMN about 18 years after a prior diagnosis of TGCT is novel. Although there is evidence showing a common clonal origin between teratoma and malignant transformation with LOH (Kum, 2012), we added to the current body of knowledge with sequencing data to support this important basic principle in germ cell tumor biology. As far as we know, there are no true teratoma cell lines derived from a pure mature teratoma, including SuSa, TERA1/2, GCT27, GCT35, GCT48, GCT44/46/72. All of these supposedly teratoma cell lines have embryonal carcinoma or yolk sac tumor derivations. In contrast, we obtained teratoma cell lines after (not before) chemotherapy, in an effort to eliminate the embryonal carcinoma and yolk sac tumor components, and performed primary cell culture on a clinically evident and pathologically confirmed pure mature teratoma for the first time in this report.

I found the manuscript confusing. While the primary objective is to demonstrate a common cell progenitor in SMN, the translational studies have been conducted using transformed teratoma, inferring that teratoma:transformed teratoma=primary GCT:SMN.

  1. The authors need to prove this correlation to use the transformed teratoma model as SMN. This assumption is not based on solid scientific evidence
  2. Several studies have demonstrated the association between active treatments, RT and chemotherapy, and development of SMN in GCT patients. To demonstrate that SMN are arising from GCT cells the authors should analyze a cohort of GCT patients who developed SMN and were not previously exposed to neither chemotehrapy or RT.

Overall, the data presented do not support the authors' conclusion that SMN originate from a common stem-cell from the original GCT.

In an effort to demonstrate a common progenitor cell in the origin of SMN with i(12p)/12p gain, we hypothesized that transformed teratoma could account for some of those SMN cases, in which the teratomas were not removed after chemotherapy. We tested this hypothesis by examining the genetic profiles of teratoma/transformed teratoma, which is known for late relapse and transformation to non-GCT lineages, in case genetic mutations caused or contributed to malignant transformation and/or SMN.

Importantly, we found that a majority of SMN had occurred in patients with primary pure seminoma, which normally does not undergo malignant transformation. Therefore, the transformed teratoma model may only explain some of those SMN cases in our study.

In order to account for most if not all SMN, we tested the hypothesis that a common (aberrant) progenitor cell that gives rise to SMN could be embedded in a primary or metastatic GCT, as well as within well-differentiated tissue, similar to mature teratoma (Kernek, 2003; Jones, 2006). It is plausible that prior radiotherapy and/or chemotherapy could have damaged the progenitor cell and initiated the malignant process. Interestingly, one patient (#14) had neither of those prior treatments. Furthermore, a majority of those SMN associated with early-staged seminoma and prior low-dose radiotherapy without chemotherapy occurred outside the radiation field (lines 169-173).

We hope that the discovery of certain progenitor stem-like cells within mature teratoma derived from distinct GCT subgroups will lay the groundwork and eventually lend support to the hypothesis that certain progenitor stem-like cells may lead to malignant transformation and to SMN with non-GCT lineages and phenotypes.

We have reworded our conclusions in order not to overstate beyond our available data.  In Conclusions, lines 285 and 287: “We discovered putative progenitor cells embedded within mature teratomas” and “We were able to recapitulate teratoma formation in one case of a xenograft cell line.”

Reviewer 2 Report

In this study Umbreit et al. characterized genomic and gene expression profiles of tumor tissues obtained from 42 survivors of TGCT, with metastatic teratoma (mTER), who developed subsequent solid malignant neoplasms (SMN) within median time of 18 years. They found the presence of additional copies of chr 12 or i(12p) by FISH in 50% of SMN. In 7 samples with mTER and areas of somatic transformation they performed further molecular analysis and found close concordance between DNA mutation configuration and gene expression profiles. They also established multipotent cell lines from two post-chemotherapy teratomas, one of which formed a teratoma after xenografting to a mouse model.  Both cell lines expressed multiple embryonic and cancer markers. The authors suggested a “common progenitor” cell of origin in SMN and TGCT.  This is an interesting and intriguing study, with potentially important implications. However, there are some points that need to be clarified:

Specific points:

  1. I have a problem with the term “common progenitor”. TGCT are considered to have a common progenitor of germ cell neoplasia in situ (GCNIS), which then progresses either to seminoma or embryonal carcinoma (EC), the latter further differentiating to teratomas or extraembryonic elements. It is also believed that seminoma can reprogram to EC and then further differentiate to teratomas and other components. The authors nicely showed that the metastatic teratoma likely contains cells that give rise to other somatic tumors, but these cannot be called “common progenitors” – it is more like a progression from microspread of EC cells, which lodge in remote places and then respond top clues of the tissue microenvironment, and develop into somatic tumors with specific histology (e.g. gastrointestinal etc. ). Please comment on that in the discussion.
  2. Table 1 is very confusing and I have several problems with it:
    1. It is is not clear which patients and tumors this table encompasses: why 20 tumours are listed while in the text the authors refer to Table 1 as showing 21 samples that were investigated by FISH? Moreover, the authors mentioned 42 tissue samples in the M&M? Why all 42 are not listed in Table 1? I think that all samples ought to be shown in Table 1 because the samples that were negative for chr. 12 amplification are as important as the positive samples.
    2. Please explain what do you mean by “primary tumor”: the paper emphasizes metastatic teratomas but this table lists mainly seminomas. Are these primary testicular tumors? Unilateral or bilateral? The table should have three histology columns: of the primary testicular tumors, metastatic tumors and subsequent somatic malignancies.
  3. It is important to discuss any discordance between TGCT and SMN concerning gain of chr. 12. From the description in the paper it is not clear.
  4. In the concluding remarks, two statements are probably too far-fetched and not proven by the data:
    1. “We discovered progenitor cells embedded within mature teratomas”. I think that the findings do not allow such statement. Unless the authors can provide pictures of sections showing IHC evidence for EC cells within teratomas.
    2. “We were able to recapitulate teratoma formation in xenograft models” – here the authors should specify that they observed teratoma formation in one case of one xenografted cell line.
  5. The two mTER-derived cell lines – apparently only a primitive, poorly differentiated cells have grown into a cell line. Did the authors analyse DNA/RNAseq profiles of the cell lines? If yes – are the profiles similar to the tumor of origin and SMN? How about the amplification of 12 /i(12p)?

Minor points:

  1. Line 189 (Discussion) the authors cited refs 23-36 while discussing microchimerism during pregnancy. 23 is an apparent typo – correct to 33-36.
  2. Ref 33: “Proc Matl Aced Sci ISA” – PNAS?

Author Response

Reviewer #2

In this study Umbreit et al. characterized genomic and gene expression profiles of tumor tissues obtained from 42 survivors of TGCT, with metastatic teratoma (mTER), who developed subsequent solid malignant neoplasms (SMN) within median time of 18 years. They found the presence of additional copies of chr 12 or i(12p) by FISH in 50% of SMN. In 7 samples with mTER and areas of somatic transformation they performed further molecular analysis and found close concordance between DNA mutation configuration and gene expression profiles. They also established multipotent cell lines from two post-chemotherapy teratomas, one of which formed a teratoma after xenografting to a mouse model.  Both cell lines expressed multiple embryonic and cancer markers. The authors suggested a “common progenitor” cell of origin in SMN and TGCT.  This is an interesting and intriguing study, with potentially important implications. However, there are some points that need to be clarified:

Thank you for your succinct summary and positive comments.

Specific points:

  1. I have a problem with the term “common progenitor”. TGCT are considered to have a common progenitor of germ cell neoplasia in situ (GCNIS), which then progresses either to seminoma or embryonal carcinoma (EC), the latter further differentiating to teratomas or extraembryonic elements. It is also believed that seminoma can reprogram to EC and then further differentiate to teratomas and other components. The authors nicely showed that the metastatic teratoma likely contains cells that give rise to other somatic tumors, but these cannot be called “common progenitors” – it is more like a progression from microspread of EC cells, which lodge in remote places and then respond top clues of the tissue microenvironment, and develop into somatic tumors with specific histology (e.g. gastrointestinal etc. ). Please comment on that in the discussion.

We agree with the reviewer that an EC-like progenitor (pluripotent) cell could differentiate into various lineages and phenotypes to engender malignant transformation and SMN. It is difficult to isolate these “micropread” EC-like progenitor cells “lodged in remote places.” Fortunately, we managed to demonstrate their existence within teratoma after chemotherapy in support of this important concept. In Discussions, we used microchimerism as an example (lines 175-185) to explain this observation. Without a better term to describe “common clonal origin”, we have replaced “common progenitor” with “progenitor stem-like” cells in the abstract and in the text (Abstract, line 38; Discussions, line 161; Conclusions, line 290).

  1. Table 1 is very confusing and I have several problems with it:
    1. It is is not clear which patients and tumors this table encompasses: why 20 tumours are listed while in the text the authors refer to Table 1 as showing 21 samples that were investigated by FISH? Moreover, the authors mentioned 42 tissue samples in the M&M? Why all 42 are not listed in Table 1? I think that all samples ought to be shown in Table 1 because the samples that were negative for chr. 12 amplification are as important as the positive samples.
    2. Please explain what do you mean by “primary tumor”: the paper emphasizes metastatic teratomas but this table lists mainly seminomas. Are these primary testicular tumors? Unilateral or bilateral? The table should have three histology columns: of the primary testicular tumors, metastatic tumors and subsequent somatic malignancies.

We agree that all 43 samples should be listed, which we initially did in Figure 1A. We have now converted this figure into Table 1, listing all cases of SMNs. The previous Table 1 is now Table 2, which we have kept separate in case that may cause confusion when we focus on and examine the nature of positive cases.

We have changed the columns in the new Table 2 from “Primary histology” to “Primary Testicular Tumor” (unilateral, unless otherwise noted in legend), “Site of SMN” to “Metastatic tumor,” and keep “SMN Histology” as it is, as recommended.

  1. It is important to discuss any discordance between TGCT and SMN concerning gain of chr. 12. From the description in the paper it is not clear.

Because about 80% of TGCT harbor i(12p) and 20% 12p gain, any discordance between the primary TGCT and SMN concerning (i[12p]/12p gain) is of importance. Interestingly, one of our original SMN cases not included in this cohort, showed i(12p) performed by FISH at one institution (MDACC) but 12p gain at another (Mayo).

In Discussions, lines 217-219 and 221-222: We alluded to and expanded on this point, “A limitation of this study relates to rarity of disease and paucity of clinical samples. We need to perform targeted DNA sequencing and RNASeq on matched patient specimens to compare primary tumors with SMNs as well as among SMNs in patients with more than one SMNs.” “However, we hope that the raised awareness from this study will spur expanded profiling on greater numbers of pertinent tumors.”  

  1. In the concluding remarks, two statements are probably too far-fetched and not proven by the data:
    1. “We discovered progenitor cells embedded within mature teratomas”. I think that the findings do not allow such statement. Unless the authors can provide pictures of sections showing IHC evidence for EC cells within teratomas.
    2. “We were able to recapitulate teratoma formation in xenograft models” – here the authors should specify that they observed teratoma formation in one case of one xenografted cell line.

Thank you for your recommendations not to overstate beyond our available data at this time.

In Conclusions, lines 285 and 287: “We discovered putative progenitor cells embedded within mature teratomas.” “We were able to recapitulate teratoma formation in one case of a xenografted cell line.”

  1. The two mTER-derived cell lines – apparently only a primitive, poorly differentiated cells have grown into a cell line. Did the authors analyse DNA/RNAseq profiles of the cell lines? If yes – are the profiles similar to the tumor of origin and SMN? How about the amplification of 12 /i(12p)?

Indeed, tera13 cell line has i(12p). However, tera21 has neither i(12p) nor 12p gain. As noted in Supplemental Table 2, tera21 was derived from patient #29 who had a residual teratoma in the pelvis after chemotherapy. However, he had no overt primary testicular tumor, although there was microlithiasis (suspect burnt-out tumor). We plan to analyze DNA/RNA seq profiles of the cell lines and tumors of origin (if available) as soon as we secure funding.   

Minor points:

  1. Line 189 (Discussion) the authors cited refs 23-36 while discussing microchimerism during pregnancy. 23 is an apparent typo – correct to 33-36.
  2. Ref 33: “Proc Matl Aced Sci ISA” – PNAS?

We have corrected the typographical errors.

Reviewer 3 Report

In my opinion the statistical analyses in reference 3 were not accurate and based on previous studies and my personal experience the statment "13% of primary nonseminomatous germ cell tumors (NSGCT) and 21% of residual metastatic NSGCT after chemotherapy" show transformation is not correct. Can the author provide other references then self citation using reference 3 for this statement? Of interest is that a previous study from the same institution reported that only 33 cases between 1985 to 2007 showed transformation which is contradicting the authors current statement.

I dont think that the statement "despite overall de-intensification of therapy" is correct. There was no "overall de-intensification" in my opinion. Please provide references if you think otherwise.

To me the introduction is to long and should be rewritten that the reader can clearly understand the background, research nice and aim of this paper. Even after several attempts reading the abstracts on several days those are not clear to me.

Why dont the authors just cite this article they mention "https://www.ncbi.nlm.nih.gov/pmc/articles/PMC4511273/" with a reference managing tool?

Author Response

Reviewer #3

In my opinion the statistical analyses in reference 3 were not accurate and based on previous studies and my personal experience the statment "13% of primary nonseminomatous germ cell tumors (NSGCT) and 21% of residual metastatic NSGCT after chemotherapy" show transformation is not correct. Can the author provide other references then self citation using reference 3 for this statement? Of interest is that a previous study from the same institution reported that only 33 cases between 1985 to 2007 showed transformation which is contradicting the authors current statement.

Thank you for pointing out an error on our part (lines 49-50): malignant transformation was observed in 15/615 (2.4%) primary NSGCT and 21/615 (3.4%) of metastatic NSGCT after chemotherapy from Jan 2000 to Dec 2010. This is compatible with a paper by Colecchia et al, in which malignant transformation was observed in 16/1,985 (0.8%) primary GCT and 24/1,985 (1.2%) metastatic GCT (However, the denominator included both NSGCT and seminoma).

Colecchia M, Necchi A, Paolini B, et al. Teratoma with somatic-type malignant components in germ cell tumors of the testis: a clinicopathologic analysis of 40 cases with outcome correlation. Int J Surg Pathol 2011;19:321-7.

I dont think that the statement "despite overall de-intensification of therapy" is correct. There was no "overall de-intensification" in my opinion. Please provide references if you think otherwise.

We stated overall de-intensification of therapy based on utilization of surgery to remove teratomas, refinement of radiation dosages to treat seminomas, establishment of active surveillance to manage stage I GCT, application of risk stratification to tailor treatments (i.e., demonstration that more cycles of BEP and higher dosage of cisplatin did not improve survival).

Jones WG, Fossa SD, Mead GM, et al. Randomized trial of 30 versus 20 Gy in the adjuvant treatment of stage I testicular seminoma: A report on Medical Research Council Trial TE18, European Organization for the Research and Treatment of cancer trial 30942. J Clin Oncol 2005;23:1200-8.

Feldman DR, Bosl GJ, Sheinfeld J, et al. Medical treatment of advanced testicular cancer. JAMA 2008;299:672-84.

Kollmannsberger C, Tyldesley S, Moore C, et al. Evolution in management of testicular seminoma: population-based outcomes with selective utilization of active therapies. Ann Oncol 2011;22:888.

Hanna N, Einhorn LH. Testicular cancer: A reflection on 50 years of discovery. J Clin Oncol 2014;32:

To me the introduction is to long and should be rewritten that the reader can clearly understand the background, research nice and aim of this paper. Even after several attempts reading the abstracts on several days those are not clear to me.

We have shortened the Introduction to make the background clearer and simpler, as recommended. For those who are interested, we have moved the additional information and details to Discussions (lines 186-193).

Why dont the authors just cite this article they mention "https://www.ncbi.nlm.nih.gov/pmc/articles/PMC4511273/" with a reference managing tool?

Thank you for suggesting addition of this reference—we have added it.  Chen K, Meric-Bernstam F, Zhao H, et al. Clinical actionability enhanced through deep targeted sequencing of solid tumors. Clin Chem 2015;61:544-53.

Reviewer 4 Report

In this article the authors explore the origin of malignancies emerging in patients with prior testicular germ cell tumor. Authors use the i(12p) as an indication of the relationship between a primary TGCT and the subsequent secondary malignancy. Also, authors compared the post-chemotherapy residual mature teratoma and the adjacent somatic transformation malignancy, showing that they are highly concordant in DNA and RNA sequencing data. They also establish teratoma-derived cell lines and xenografts which showed some evidence of stemness-related markers.

The work is interesting and expands on the knowledge about this topic. It is well written (although some minor typos should be corrected).

I have the following suggestions:

  • Authors should better clarify what they mean by “normal control tissues” where they performed the i(12p)
  • The resolution of FISH images should be improved, since they are very blurred in the pdf.
  • Regarding the cell cultures of teratoma and the controls with fibrosis/necrosis, were these specimens confirmed by a dedicated Pathologist, to confirm if no other elements, namely viable tumor, was present?
  • How do authors interpret the mutations present in all cases of paired teratoma and somatic transformation in genes like TP53, NOTCH2, MLL3, MAP3K4?
  • Authors should explore more the genes for which they found differences in differential expression between teratoma and not teratoma – what is the role of these genes and what could be the meaning of this differential expression?

Author Response

Reviewer #4

In this article the authors explore the origin of malignancies emerging in patients with prior testicular germ cell tumor. Authors use the i(12p) as an indication of the relationship between a primary TGCT and the subsequent secondary malignancy. Also, authors compared the post-chemotherapy residual mature teratoma and the adjacent somatic transformation malignancy, showing that they are highly concordant in DNA and RNA sequencing data. They also establish teratoma-derived cell lines and xenografts which showed some evidence of stemness-related markers.

The work is interesting and expands on the knowledge about this topic. It is well written (although some minor typos should be corrected).

Thank you for your succinct summary and positive comments. We have corrected those minor typos in lines 180 and 367 and in Supplemental Table 2.

I have the following suggestions:

Authors should better clarify what they mean by “normal control tissues” where they performed the i(12p)

In Results, line 85: “Two separate patients with SMN positive for i(12p) underwent testing of normal adjacent tissue without cancer as negative control.”

The resolution of FISH images should be improved, since they are very blurred in the pdf.

Regarding the cell cultures of teratoma and the controls with fibrosis/necrosis, were these specimens confirmed by a dedicated Pathologist, to confirm if no other elements, namely viable tumor, was present?

We have provided higher resolution images in Figure 1, as requested.

 How do authors interpret the mutations present in all cases of paired teratoma and somatic transformation in genes like TP53, NOTCH2, MLL3, MAP3K4?

In Discussion, lines 198-201: “We found that metastatic teratomas and their corresponding somatic transformations displayed highly concordant genetic profiles based on genome sequencing, which points to a common clonal origin and suggests that specific genetic aberrations are not responsible for dedifferentiation or somatic transformation.”

Authors should explore more the genes for which they found differences in differential expression between teratoma and not teratoma – what is the role of these genes and what could be the meaning of this differential expression?

We agree. When mutations occur rarely and randomly in either teratoma and/or non-teratoma components within the same tumor, it is difficult to assign a role or meaning (e.g., driver vs passenger) for their presence. For example, we detected FLT1 mutations in a non-teratoma (3 patients), a teratoma (1 patient), and both non-teratoma and teratoma (1 patient)(Figure 2). Although there are purportedly specific mutations for PNET (patient 23), such as FLI 1, or for rhabdomyosarcoma (patient 26), such as PAX3 or PAX7 (Supplemental Table 1), we did not detect them in our patients. However, we agree that we now have a unique opportunity to explore their role and meaning when we examine such mutations in additional cases.  

Round 2

Reviewer 1 Report

The authors have responded to my comments and requests.

Reviewer 4 Report

No further suggestions to the authors.